# Oleaginous Yeast Extracts and Their Possible Effects on Human Health

**DOI:** 10.3390/microorganisms11020492

**Published:** 2023-02-16

**Authors:** Marie Vysoka, Martin Szotkowski, Eva Slaninova, Lucia Dzuricka, Paulina Strecanska, Jana Blazkova, Ivana Marova

**Affiliations:** Faculty of Chemistry, Brno University of Technology, Purkynova 118, 61200 Brno, Czech Republic

**Keywords:** oleaginous yeast, pigmented yeasts, bioactive compounds, antimicrobial effect, cytotoxic effect, apoptosis

## Abstract

Four non-conventional oleaginous and pigmented yeast strains of *Metschnikowia pulcherrima*, *Cystofilobasidium infirmominiatum*, *Phaffia rhodozyma*, and *Rhodotorula kratochvilovae* were used in this study. Complex yeast extracts were prepared and tested for biological activity, safety, and effect on human health. In this paper, we measured the antioxidant activity and antimicrobial effect of yeast biomass as a whole and their extracts to compare the influence of carotenoids and other bioactive substances in the studied biomass. All yeast extracts exhibited a significant dose-dependent antimicrobial effect against both G+ and G- bacteria and had a strong antioxidant effect. No cytotoxicity in the mouse melanoma B16F1 cell line was found in concentrations up to 20% of rehydrated biomass in cell medium. All of the extracts were cytotoxic at a concentration of 5 mg of extract/g of dry biomass. All the pigmented yeast extracts showed some positive results for apoptosis of murine melanoma cell lines and are therefore strong candidates positively effect human health. Red yeast cell biomass is a prospective material with many attractive biological functions and can be used in the food industry, as a pharmaceutical material, or in the feed industry.

## 1. Introduction

The use of “natural” or alternative medicines has increased markedly over the last few years. More and more older adults (i.e., baby boomers) are using complementary and alternative medicine—dietary supplements and herbal remedies—without advice from a physician on the assumption that these substances will have a beneficial effect. Health benefits are the most important uptake for this research.

It has been well documented that yeasts have many applications in fermentation, food, feed, agricultural, biofuel, medical, and chemical industries, as well as environmental protection [1]. They have been used for the production of fermented food for at least as long as 7000 BC [2]. Yeast biomass from so-called nutritional yeasts is widely used as a source of nutritional components, such as single-cell protein (SCP) [3,4]. Dried and killed *Y. lipolytica* protein-rich biomass is recognized as safe for human and animal nutrition in accordance with current food and nutrition safety laws. It is worth emphasizing that *Y. lipolytica* has not caused any allergic reactions in humans. It is a valuable source of bioactive compounds such as proteins, trace minerals, vitamins, and other valuable compounds such as yeast biomass itself [5]. The occasional occurrence of opportunistic infections of *Y. lipolytica* in immunocompromised and catheterized patients does not differ from other microorganisms with a history of safe use, such as *S. cerevisiae* [6]. The natural occurrence of *Y. lipolytica* in foods, particularly cheese [7], other dairy products [8], meat [9], fish [10], and also some soft drinks [11], is an additional argument in favor of its safety. All these examples suggest that *Y. lipolytica* may be a regular, although probably minor, component of food microbiota and can also serve as probiotics and prebiotics [6].

Microbial food cultures have directly or indirectly come under various regulatory frameworks in the course of the last decades. Several of those regulatory frameworks put emphasis on “the history of use”, “traditional food”, or “general recognition of safety”. Authoritative lists of microorganisms with documented uses in food have therefore come into high demand. One such list was published in 2002 as a result of a joint project between the International Dairy Federation (IDF) and the European Food and Feed Cultures Association (EFFCA). The “2002 IDF inventory” has become a de facto reference for food cultures in practical use. However, as the focus was mainly on commercially available dairy cultures, there was an unmet need for a list with a wider scope. A recent study presents an updated inventory of microorganisms used in food fermentations covering a wide range of food matrices (dairy, meat, fish, vegetables, legumes, cereals, beverages, and vinegar) [12]. To this list were added not only the yeast *Yarrowia lipolytica* but also *Cystofilobasidium infirmominiatum* (dairy) and *Metchnikowia pulcherrima* (wine) [12]. *Phaffia rhodozyma* and *Rhodotorula mucilaginosa* belong to very similar oleaginous pigmented yeasts present in several foods and beverages (peanuts, apple cider, cherries, fresh fruits, fruit juice, cheese, sausages, and edible molluscs [13]), and similar to *Y.lipolytica* have broad biotechnological potential [14,15]. 

*Rhodotorula* yeasts produce pink to red colonies and blastoconidia that are unicellular and lacking pseudohyphae and hyphae. Several authors have isolated *Rhodotorula* in different ecosystems and environments. Previously considered nonpathogenic, *Rhodotorula* species have emerged as opportunistic pathogens with the ability to form colonies and infect susceptible patients. Recent studies have demonstrated that the incidence of fungemia caused by *Rhodotorula* was between 0.5% and 2.3% in the USA and Europe, respectively. Most cases of infection with *Rhodotorula* fungemia are associated with central catheters in patients with hematologic malignancies. To make an updated and critical evaluation of red yeast safety, we have focused this study on the analysis of cytotoxicity on human cell lines, apotosis, and potential benefits of freeze-dried, immortalized pigmented yeast biomass [13].

In general, yeast biomass contains valuable fatty acids, carbohydrates, nucleic acids, vitamins, and minerals. It is rich in certain essential amino acids, such as lysine and methionine, which are limited in most plant and animal foods [1,14]. Some of them, so-called “red” yeasts, can produce carotenoid pigments. *Phaffia rhodozyma*, another red yeast, produces in large quantities the carotenoid pigment astaxanthin, but this yeast also has the ability to produce torulene and torularhodin in its metabolic pathway. These pigments are also produced by the above-mentioned pigmented yeasts of the genera *Cystofilobasidium* and *Rhodotorula*, while the oleaginous yeast *Metchnikowia pulcherima* produces the antimicrobial pigment pulcherrimin [12,13,14,15]. Torularhodin and torulene are two widespread microbial carotenoids. Torularhodin shows considerable antioxidant activity that helps the stabilization of membranes under stress conditions. These carotenoids are beneficial because they are precursors of vitamin A and hormones, and they have anti-aging and antioxidant capacities. They may also prevent certain types of cancer and enhance the immune system. These possibilities make torularhodin and torulene a hot research topic in carotenoid biotechnology [15].

In this paper, we have evaluated some biological effects such as antioxidant activity and the antimicrobial effect of yeast biomass as a whole and their extracts to compare the influence of carotenoids, glucan, lipids, and other bioactive substances in the studied biomass. Four different strains of pigmented oleaginous yeasts recognized as “safe” [12] were enrolled in the comparative study. The safety of yeast extracts will be verified by an MTT test using mouse melanoma cells. The main conclusion of this study was to evaluate the beneficial effect of yeast biomass on human health and its possible anticancer effect. 

## 2. Materials and Methods

### 2.1. Cultivation of Microbial Cells and Their Processing

Yeast strains of *Metschnikowia pulcherrima* CCY 029-002-145, *Cystofilobasidium infirmominiatum* CCY 17-18-4, *Phaffia rhodozyma* CCY 77-1, and *Rhodotorula kratochvilovae* CCY 20-2-26 were obtained from the Culture Collection of Yeast (Institute of Chemistry, Slovak Academy of Science, Bratislava, Slovakia). The selection of yeast strains was based on a previous publication focused on a screening study of all producers [13]. In this study some of produced metabolites (such as carotenoids, lipids and PUFA) with potential biological effect on human cells were quantified. 

First, the yeast from frozen stock was cultured on YPD (yeast extract peptone dextrose) agar plates for 72 h at laboratory temperature (24 °C). Next, the inoculum was prepared by transferring one biological loop from an agar plate into 100 mL of sterile YPD medium (yeast extract, 10 g·L^−1^; peptone, 20 g·L^−1^, glucose, 20 g·L^−1^) in an Erlenmeyer flask (250 mL) and cultured under a constant shaking regime (120 rpm) for 24 h. The yeasts were then inoculated directly into the production medium in a ratio of 1:5 (inoculum: production medium) and cultivated for 96 h at laboratory temperature. The production medium is composed of glucose 69.29 g·L^−1^, KNO_3_ 1.52 g·L^−1^, KH_2_PO_4_ 4 g·L^−1^, and MgSO_4_∙7H_2_O 0.7 g·L^−1^. The production media were chosen according to the previous research [16]. Before inoculation, each medium was sterilized for 45 min at 120 °C and then cooled to room temperature. Biomass was washed three times with distilled water and then freeze-dried. Biomass that was used for microbial tests and tests on B16F1 cell lines was first immortalized by exposure to high temperatures for 10 min.

### 2.2. Preparation of Microbial Extracts

#### 2.2.1. Rehydrated Biomass

15 ± 5 mg of freeze-dried biomass was weight and rehydrated by addition of 1 mL distilled water in plastic extraction tubes.

#### 2.2.2. Folch Extract Preparation 

15 ± 5 mg of freeze-dried biomass was weight and rehydrated by addition of 1 mL distilled water in plastic extraction tubes. The water was removed by centrifugation and 250 ± 50 mg acid-washed glass beads (250–500 μm diameter, Roth, Germany) and 1 mL of methanol were added to the pellet. To obtain biomass with ruptured cell walls for analysis, it was necessary to vortex for 10 min. The content of the PP tube was transferred into a glass reaction tube by washing it with 2000 μL of chloroform, and the glass tube was vortexed for 10 min. Then, 1 mL of distilled water was added for the phase separation. After centrifugation (3000 rpm/5 min/4 °C), the separated bottom chloroform phase with extracted pigments was evaporated under nitrogen at 25 °C, followed by the addition of 1 mL of an ethylacetate:acetonitrile (1:3) mixture. The ethylacetate:acetonitrile mixture containing extracted pigments was filtered through a syringe filter (0.45 µm, PTFE membrane, 13 mm) into 2 mL vials.

### 2.3. Characterization of Yeast Extracts

#### 2.3.1. Antioxidant Activity

ABTS was dissolved in distilled water at a concentration of 7 mmol/L. Radical anion ABTS + was obtained through a reaction with 2.45 mmol/L potassium persulphate. The solution was left in the dark for 12 h at room temperature. Before the measurement, the ABTS + was diluted in ethanol to obtain an absorbance of about A = 0.7 and λ = 734 nm. Then the reaction starts with 1 mL of ABTS + and 10 μL of rehydrated biomass extract. Absorbance was measured at 0 and 10 min. As a blank, distilled water was used. 

For calibration a Trolox solution in the concentration range of 50–400 μg/mL was prepared.
Calibration curve equation: *y* = 15,388 · *x* mg/mL

#### 2.3.2. Antimicrobial Activity

Antimicrobial activity is tested using standard antimicrobial assays. The antimicrobial efficacy of the test substances and extracts was monitored against gram-positive strains—*Micrococcus luteus* CCM 1659—and gram-negative strains—*Serratia marcescens* CCM 303. Microorganisms were obtained from the Czech Collection of Microorganisms (CCM, Prague, Czech Republic). Lyophilized cultures were rehydrated and cultured according to manufacturer recommendations. Both microorganisms were cultivated on BHI media (Himeda, Czech Republic) at 37 °C for 24 h. Microorganisms were subsequently seeded into fresh media at a concentration of 0.5 McFarland units (approximately 1.5·10^8^ cells/mL). In the case of cultivation on solid media, 20 g/L of microbiological agar (Sigma-Aldrich, St. Louis, MI, USA) was added, prior to the sterilization, to the media.

For the purpose of this study, preliminary broth dilution tests in a 96-well plate were carried out to determine whether rehydrated biomass and yeast extracts possess good antimicrobial activity. Moreover, disc diffusion tests were performed in some cases.

#### 2.3.3. Methods of Analysis of Yeast Metabolites

##### HPLC Analysis of Carotenoids, Sterols, and Ubiquinone

The frozen biomass was lyophilized and subjected to gravimetric determination of biomass production (g/L). The hydrated biomass in microtubes filled with glass beads and methanol was subsequently subjected to a disintegration process using a laboratory disintegrator for the purpose of extracting pigments, ergosterol, and ubiquinone according to Folch’s extraction. Quantitative analysis was performed using calibration standards (beta-carotene, astaxanthin, lycopene, lutein, violaxanthin, neoxanthin, and ergosterol) from the regress equation. The following gradient elutions were used: Mobile phases A (Acetonitrile 840 mL Methanol 20 mL 0.1 M Tris-HCl (pH = 8) 140 mL), B (Methanol 680 mL Ethyl acetate 320 mL). The final extract was stripped of the extraction solvent (chloroform) and dissolved in a 2:1 HPLC-grade solvent mixture of ethyl acetate and acetonitrile in a volume suitable for HPLC analysis with PDA detection according to [17].

##### Analysis of Lipids and Fatty Acids

20 mg of freeze-dried biomass was weighed into crimp vials. Subsequently, 1.8 mL of a transesterification mixture containing 0.5 mg/mL of dissolved C17:0 internal standard (heptadecanoic acid, Merck) in 15% sulfuric acid in methanol was added. The vials were heated to 85 °C for 2 h. After cooling, the entire content of the vial was transferred to a 4 mL vial with 0.5 mL of 0.05 M NaOH. 1 mL of HPLC hexane was pipetted into the sample. The resulting mixture was then vortexed vigorously for 5 min. After phase separation, 100 μL of the upper hexane phase was collected in a glass vial for HPLC/GC with 900 μL of pre-pipetted HPLC hexane. According to [18], samples with fatty acid methyl esters were then analyzed by a gas chromatograph Thermo Scientific TRACETM Gas Chromatograph with a Thermo Scientific A1 1310 autosampler, Zebron ZB-FAME column (30 m × 0.25 mm × 0.25 μm) and a flame ionization detector (FID).

Determined fatty acids were divided into three groups: polyunsaturated fatty acids (PUFA), monounsaturated fatty acids (MUFA), and saturated fatty acids (SFA). The results were graphically evaluated by showing the individual ratios of PUFA, MUFA, and SFA in the total amount of fatty acids in the biomass. The total percentage of fatty acids in the biomass feed was also evaluated.

##### Analysis of Other Metabolites

Beta-Glucans were analyzed by an enzyme kit (Megazyme ELISA Yeast Glucans Kit) according to [19].

### 2.4. Cytotoxicity Tests/Interaction with Human Cells

#### 2.4.1. Cell Cultures

B16F1 cell lines were cultivated in DMEM (Dulbecco’s modified Eagle medium), high glucose medium with 1% of antibiotics and 10% of FBS (Fetal Bovine Serum), at 37 °C and 5% of CO_2_ atmosphere.

#### 2.4.2. Tetrazolium Blue Assay-MTT Test 

The MTT assay is the first high-throughput screening cell viability assay developed for a 96-well format. Solid materials are extracted in cell culture medium and multiple dilutions of the extract are prepared and added into each well containing 1 × 10^4^ cells. After 24 h incubation, the water-soluble MTT substrate (yellowish solution when prepared in media or salt solutions lacking phenol red) is added and incubated with Raji cells for 3 h, and then MTT is converted to an insoluble formazan. Since formazan must be solubilized prior to absorbance readings, methods have been developed to solubilize the formazan product, stabilize the colour, avoid evaporation, and reduce interference by phenol red and other culture medium components. The purple formazan can be quantified at 540 nm by a spectrophotometer. Cytotoxicity is calculated based on the formazan signal generated, which has been shown to have good linearity up to 10^6^ cells per well and is dependent on the MTT concentration, the incubation time, and metabolically active viable cells. If the mean viability of cells is reduced to < 70% of the blank control, the sample is considered potentially cytotoxic. The exact mechanism of MTT reduction is not well understood, but it is believed to involve NADH or similar reducing molecules that transfer electrons to MTT.

Procedure:DAY: seed 1 × 10^4^ cell/100µL/1 well in a 96 well plate, incubate for 24 hDAY: add 100 µL of extract, mixed with medium for a specific concentration DAY: add 20 µL of MTT (concentration 2.5 mg/mL PBS)After 3 h add the stop solution, which is 10% SDS (sodium dodecyl sulphate) in PBS (10 g/100 mL PBS), 100µL in each well, and leave it in the dark at room temperature for 20 h.DAY: read the plates on Elisa reader at 540 λ = nm

The protocol was transformed according to [20].

#### 2.4.3. Apoptosis

Harvested cells intended for analysis were separated by centrifugation (different cells may need different centrifugation conditions), and the supernatant was discarded. Resuspended cell pellets in cold PBS were washed by gentle shaking or by up-and down mixing with a pipette tip. Re-centrifuged cells were washed again, and the supernatant was discarded.

The cell pellet was resuspended in 1× Annexin V Binding Buffer and the cell density was adjusted to 2–5 × 10^5^ cells/mL, resulting in a sufficient volume of cell suspension (100 μL per assay).

Further, 5 μL of Annexin V–FITC and 5 μL of Propidium Iodide was added to each 100 μL of cell suspension and mixed gently. Incubation was done for 15 min at room temperature in the dark. Cells were centrifuged, and the pellet was re-suspended in 100 μL of 1× Annexin V Binding Buffer. The stained cells were analysed by flow cytometry as soon as possible.

#### 2.4.4. Statistical Analysis

Data were statistically analyzed using the *Statitsica 2.0* software.

## 3. Results

### 3.1. Growth and Production of Biomass

The biomass production yield was similar in all tested strains, in the range of 13.8–15 g/L (Figure 1). The biomass yield of *S.cerevisiae* was included into this graph solely for comparison of the studied strains with well-known strains described in literature [12]. 

### 3.2. Bioactive Compounds in Biomass

Carotenoids are an essential group of compounds produced by red yeasts. They are important antioxidants, food colorants, cosmetic ingredients, and feed additives. The general composition of red yeast biomass is introduced in Table 1 [16,17,18,19]. The carotenoid content in the presented carotenogenic yeast strains *Rhodotorula kratochvilovae* and *Phafia rhodozyma* is around 1.3 mg/g of dry biomass. The main carotenoids occurring in yeast biomass are beta-carotene, torulene, and a small amount of lycopene (Table 2).

From the Figure 2, it is obvious that there is a big difference in the amounts of individual ratios between the yeasts that were used in this research. The yeast strain *Metschnikowia pulcherrima* has the highest amount of monounsaturated fatty acids. The content of polyunsaturated fatty acids differed according to strain and was in the range of 22–44% of total fatty acids.

Figure 3 shows the percentage of individual fatty acids in the strain *Rhodotorula kratochvilovae*. Significantly high levels of oleic acid (33.99%), palmitic acid (28.50%), myristic acid (19.59%), and α-linolenoic acid were found in yeast lipids. Distribution of individual fatty acids was also measured by gas chromatography. The yeast strain *Metschnikowia pulcherrima* has the highest amount of oleic, palmitic, stearic, and linoleic acid out of the analyzed strains. 

### 3.3. Antioxidant and Antimicrobial Activity

According to the results in Table 3, we can compare the antioxidant activity of rehydrated freeze-dried biomass and of Folch extract. All samples showed high potential to be strong antioxidants. 

The antibacterial effect of red yeast lysates was tested using two model bacterial strains: gram-negative *Serratia marcescens* and gram-positive *Micrococcus luteus*. The results obtained are demonstrated in Figure 4 and Figure 5. 

According to the results of the test, the extract of *Phaffia rhodozyma* had the highest antibacterial potential at a concentration of 0.1 mg/mL and 27% of inhibition and 23% inhibition at 0.00156 mg/mL. On the other hand, the *Metschnikowia pulcherrima* yeast strain was only about 12% effective against gram-negative bacteria.

According to the test, the highest potential against *M. luteus* was shown by extracts of *Phaffia rhodozyma,* where at a concentration of 0.1 mg/mL there was 23% inhibition and at 0.0016 mg/mL the inhibition was 12%. On the other hand, the yeast strain *Rhodotorula kratochvilovae* was effective against gram-positive bacterial growth only at a concentration of 12%.

### 3.4. Influence of Yeast Extracts on the Viability of Human Cell Lines

The MTT assay was performed to evaluate the cytotoxic effect of yeast extracts on B16F1 cell lines. B16F1 cells were treated with individual yeast extracts in a concentration range of 0.04–0.28 mg/mL for 24 h. A considerable reduction in the cell viability observed was lysate concentration-dependent when compared with the control, untreated cells (Figure 6). 

In Figure 6 we see that toxicity of the yeast extracts was concentration-dependent in all concentration ranges. The viability of mouse melanoma cells with microbial extracts at a lower concentration (about 4%), which corresponds to 4 mg of dry lyophilized biomass diluted in 1 mL DMEM media, was 100% or slightly above. All extracts were at the limit of toxicity (50% viability of human cells) at the 20% concentration of yeast lysate, which corresponds to 20 mg of dry lyophilized biomass diluted in 1 mL of DMEM media. 

### 3.5. Apoptosis Testing

To evaluate the possibility of an apoptotic effect on the B16F1 cell line, it was necessary to prepare organic extracts of dry lyophilized biomass. Biomass was rehydrated, and then Folch extraction was used to extract all active compounds from the biomass. The organic solvent was evaporated under nitrogen and then dissolved in 1 mL of DMSO (dimethyl sulfoxide). Before testing, the toxicity of the sample was verified in DMEM media. The highest acceptable concentration of DMSO for cell lines was 5%. 

In Figure 7, it can be seen that the cytotoxic effects of all of the extracts were concentration-dependent, especially for the 0.31 mg/g of dry biomass. The highest cytotoxic effect was found in *C. informiniatum* CCY 17-18-4 (approx. 1 mg/g caused a 50% decrease in cell viability), while the other strains were safe at 3–4 mg/g depending on the strain type. All of the extracts were cytotoxic at a concentration of 5 mg of extract/g of dry biomass. ThAe blank sample of 5% DMSO was also tested to avoid false positive results.

From the data obtained from flow cytometry (Figure 8), it is clear that there is a correlation between cell cytotoxicity and apoptosis (Figure 9). All of the microbial extracts showed some positive results for apoptosis. From these data, we concluded that the cytotoxicity of microbial extracts was dose dependent in B16F1 cells. The toxic effects of plant extracts can be correlated to their antioxidant and antimicrobial activity.

## 4. Discussion

Red yeasts belong to non-conventional yeasts and exhibit many interesting properties. Yeast biomass contains valuable bioactive compounds—fatty acids, carbohydrates, nucleic acids, vitamins, and minerals. It is rich in certain essential amino acids, such as lysine and methionine, which are limited in most plant and animal foods (Table 1). Further, many antioxidants, provitamins, and quinones are part of the red yeast biomass [12,13,14,15,16]. 

The toxic effects of microbial extracts can be correlated to their antioxidant and antimicrobial activity. Both antioxidant and antibacterial effects were confirmed in all yeast cell lysates (Part 3.3). The antibacterial effect of red yeast lysates was tested using two model bacterial strains: gram-negative *Serratia marcescens* and gram-positive *Micrococcus luteus*. The highest antibacterial potential against gram-negative bacteria was shown by extracts of *Phaffia rhodozyma,* where at 0.1 mg/mL there was 27% inhibition and at 0.002 mg/mL there was 23% inhibition. On the other hand, the *Metschnikowia pulcherrima* yeast strain was not very effective against gram-negative bacteria. The highest potential against *M. luteus* showed also extract of *Phaffia rhodozyma,* where at 0.1 mg/mL we have found 23% of inhibition and at 0.002 mg/mL was inhibition effect still to 12%. On the other hand, the yeast strain *Rhodotorula kratochvilovae* was less effective (12%) against the gram-positive bacterium *M. luteus*.

The safety of red yeast cell lysates was verified by the MTT test (Part 3.5). The toxicity of the pigmented yeast extracts was concentration-dependent in all concentration ranges. The viability of mouse melanoma cells incubated with pigmented yeast extracts at a lower concentration (about 4%) was 100% or slightly above. Concentrations of yeast lysates up to 20% were under the limit of toxicity (50% viability of human cells [20]).

In most cases, cancer development and progression are mediated by the suppression of apoptosis. PI/Annexin V detects cellular apoptosis as a consequence of differential nuclear staining. The normal and early apoptotic cells are characterized by intact membranes; the plasma membrane undergoes structural changes that include the translocation of phosphatidylserine from the inner to the outer leaflet (extracellular side) of the plasma membrane. It has been reported that the translocated phosphatidylserine on the outer surface of the cell marks the cell for recognition and phagocytosis by macrophages [21]. 

Positive biological effects of pigmented yeast biomass can be related to the content of valuable fatty acids, sterols, beta-glucans and other carbohydrates, nucleic acids, vitamins and provitamins, minerals, quinones, certain essential amino acids, and other components [1,14]. Red yeasts can produce carotenoid pigments, such as astaxanthin, torulene, and torularhodin, all of which show considerable antioxidant and antibacterial activity and may also prevent certain types of cancer and enhance the immune system [12,13,14,15,16].

Another component of oleaginous yeast biomass are unsaturated fatty acids, with a high portion of oleic acid and up to 40% of PUFA (Figure 2). Recently, it was found that both oleic acid and alpha-linoleic acid significantly down-regulated cell proliferation, adhesion, and/or migration. In addition, it was observed that both of these fatty acids positively cross-regulate the expression levels of the AMPK/S6 axis. Moreover, they up-regulated tumor suppressor genes (p53, p21, and p27), which have an important role in esophageal cancer and, thus, might be useful agents in the management or chemoprevention of esophageal cancer [22]. 

Recent evidence suggests that patients’ nutritional status plays a major role in immunotherapy outcomes. Fatty acids are essential in a balanced diet and have been shown to influence the immune response. Moreover, short-chain fatty acids (SCFAs) show beneficial effects in metabolic disorders as well as in cancer, and polyunsaturated fatty acids (PUFAs) contribute to body weight and fat-free mass preservation in cancer patients. In line with these data, several studies imply a role for SCFAs and PUFAs in boosting the outcome of immunotherapy. The potential roles of omega-9 fatty acids in inflammation and cancer management were discussed. Preclinical and clinical evidence indicating that SCFAs and PUFAs may have the potential to boost immunotherapy efficacy was demonstrated, and opportunities for successful application of nutritional interventions focusing on SCFAs and PUFAs to increase the therapeutic potential of immunotherapeutic approaches for cancer were addressed as well [23,24]. 

Based on the results of the present study, it can be concluded that oleaginous pigmented yeast cells and cellular extracts have beneficial effects in the fight against bacteria and also have a strong antioxidant effect. Products from biomass are especially intended for people at risk of vitamin B deficiency who avoid animal products, including vegans and vegetarians, and those with a low intake of animal foods, such as populations who do not consume animal products due to culture, conviction, or restrictive diet patterns, as well as athletes and people after recovery. The safety of these lysates was confirmed by cytotoxicity tests at high concentrations. The influence of red yeast lysates on apoptosis could be taken into account in the prevention or therapy of some cancers.

Oleaginous pigmented yeast cell biomass is a prospective material with many attractive biological functions and can be used in the food industry, as a pharmaceutical material, or in the feed industry.

## Figures and Tables

**Figure 1 microorganisms-11-00492-f001:**
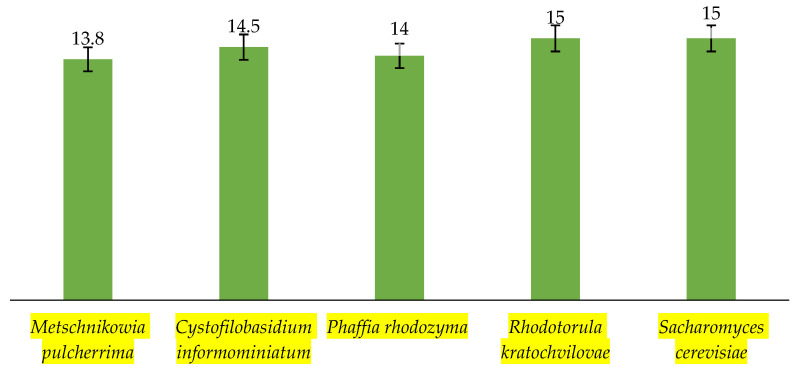
Biomass yield (g/L) after cultivation in Erlenmeyer flasks after 96 h.

**Figure 2 microorganisms-11-00492-f002:**
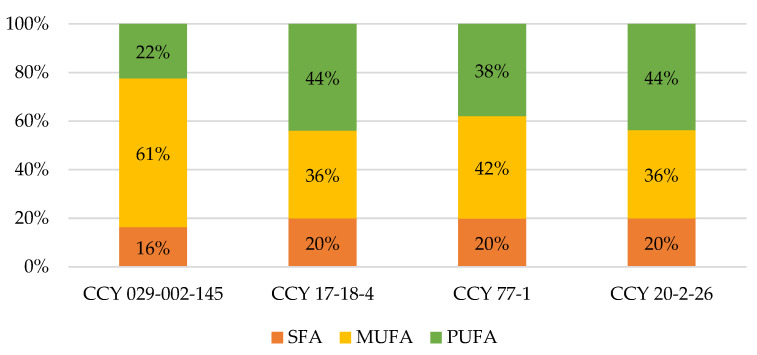
Production of fatty acids by studied pigmented yeasts.

**Figure 3 microorganisms-11-00492-f003:**
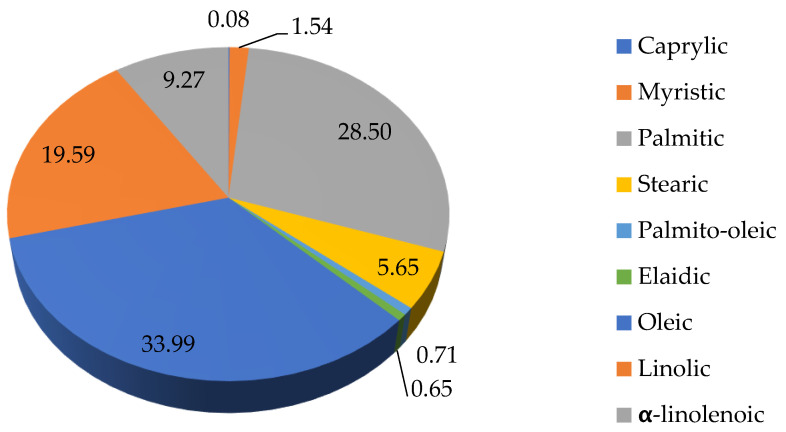
Percentage composition of various fatty acids in strain *Rhodotorula kratochvilovae*.

**Figure 4 microorganisms-11-00492-f004:**
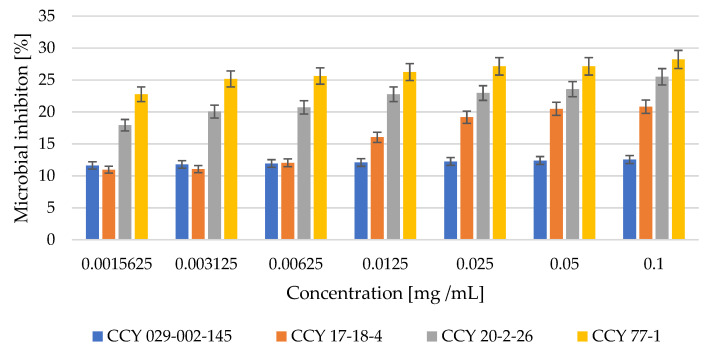
Inhibition of *Serratia marcescens* cells by yeast lysate [%].

**Figure 5 microorganisms-11-00492-f005:**
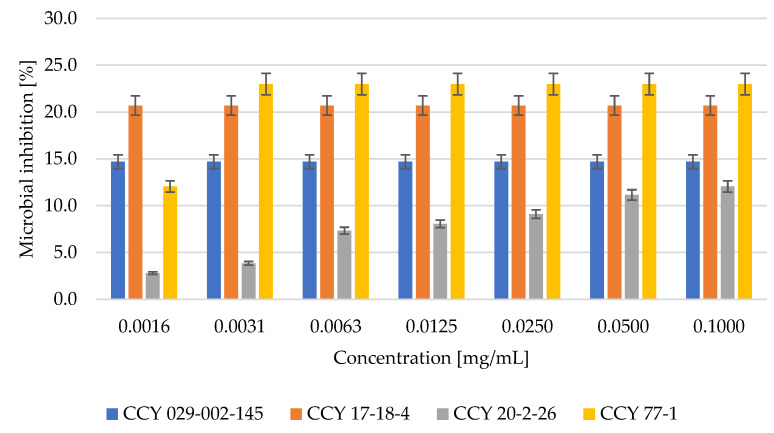
Inhibition of *Micrococcus luteus* cells by yeast lysate [%].

**Figure 6 microorganisms-11-00492-f006:**
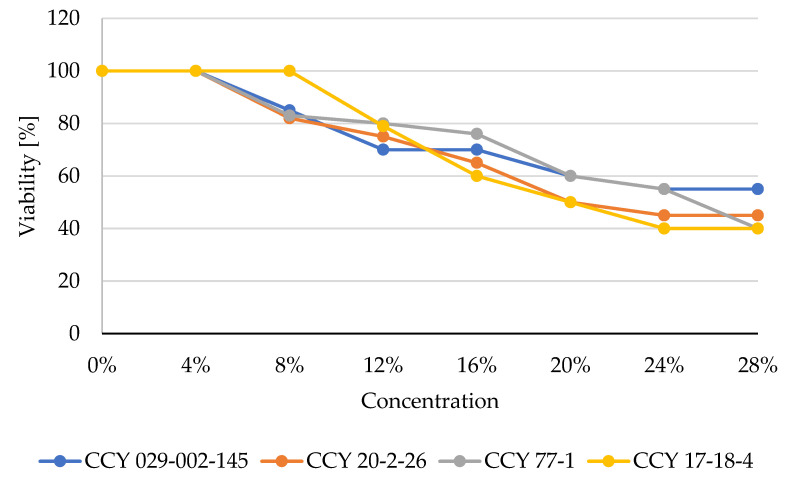
Cytotoxic effect of rehydrated yeast biomass in the concentration range of 4–28%.

**Figure 7 microorganisms-11-00492-f007:**
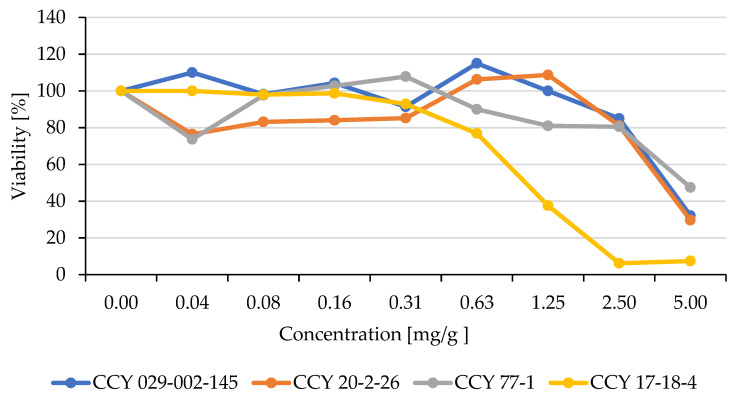
Cytotoxic effect of organic extracts from yeast biomass in the concentration range of 0.04–5 mg/g.

**Figure 8 microorganisms-11-00492-f008:**
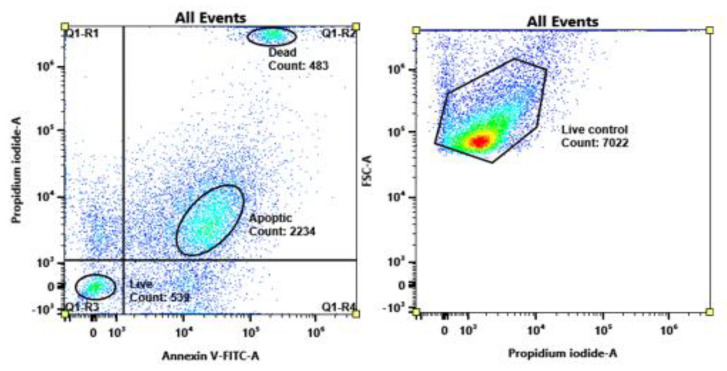
Control samples of dead, apoptotic (treated with camptothecin), and control cells of B16F1 murine melanoma cells.

**Figure 9 microorganisms-11-00492-f009:**
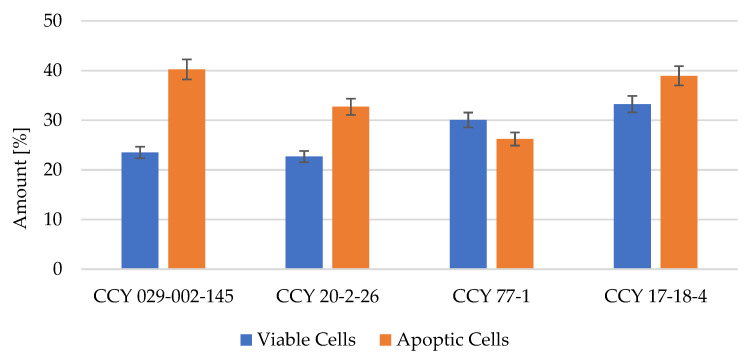
Ratio of viable and apoptotic cells after microbial extract treatment.

**Table 1 microorganisms-11-00492-t001:** Typical composition of red yeast biomass (*Rhodotorula kratochvilovae*) [11,12,13].

Parameter	Vital Biomass	Imortalized Biomass
Proteins in dry biomass (mg/g)	250–338	247–345
Glucans in dry biomass (mg/g)	110–245	118–220
Lipids in dry biomass (mg/g)	124–598	143–589
Beta-carotene (mg/g d.w.)	0.4–2.10	0.90–4.25
Total carotenoids (mg/g d.w.)	0.61–3.90	1.07–5.42
Ubiquinone (mg/g d.w.)	0.59–3.10	0.80–2.92
Ergosterol (mg/g d.w.)	0.22–3.25	0.42–3.87

**Table 2 microorganisms-11-00492-t002:** Content of carotenoids in biomass [mg/g].

	*Metschnikowia Pulcherrima*	*Cystofilobasidium* *Infirmominiatum*	*Phaffia Rhodozyma*	*Rhodotorula Kratochvilovae*
Ergosterol	0.210 ± 0.010	0.940 ± 0.012	1.121 ± 0.013	2.955 ± 0.012
Ubiquinon	0.469 ± 0.020	1.350 ± 0.021	1.548 ± 0.009	2.335 ± 0.013
Torularhodin	-	-	0.856 ± 0.009	1.063 ± 0.009
Lycopen	-	-	-	0.051 ± 0.001
Torulen	-	-	0.058 ± 0.002	0.103 ± 0.002
β- carotene	-	0.006 ± 0.001	0.024 ± 0.001	0.066 ± 0.001
Total carotenoids	-	0.007 ± 0.001	1.250 ± 0.004	1.374 ± 0.008

**Table 3 microorganisms-11-00492-t003:** Antioxidant activity of biomass.

	Antioxidant Activity [mg of Trolox Equivalent/g]	
	Rehydrated biomass	Folch extract
CCY 029-002-145	4.435 ± 0.05	4.107 ± 0.01
CCY 20-2-26	5.807 ± 0.08	5.002 ± 0.02
CCY 77-1	4.215 ± 0.02	4.321 ± 0.03
CCY 17-18-4	4.536 ± 0.07	4.612 ± 0.01

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
