# Peer review of "Oleaginous Yeast Extracts and Their Possible Effects on Human Health"

_microorganisms, 2023, doi:10.3390/microorganisms11020492_

Round 1

Reviewer 1 Report

Manuscript title: Oleaginous yeast extracts and their possible effects on human health

Major comments:

The authors strongly encouraged clarifying the identified strains, I have doubts about whether the implementation of the work has led to meaningful results.

The following are my comments and critique

Minor comments:

# As a non-native speaker, I found the manuscript easy to read and understand. However, the English writing of this paper is not acceptable and should be improved. There are some grammatical errors, and in some instances, the phrasing needs to improve also.

# Abstract: antimicrobial and antibacterial effect? only antimicrobial is enough

# line 14 “I” ?

# Abstract should support quantitative values

# Introduction and discussion should be completely improved

# Define abbreviations upon the first appearance in the text, especially scientific names

# Normalize the writing style ml/mL, minutes/min, hours/h, etc…

# Absence of culture conditions details.

# Line 81, each medium was sterilized for 45 minutes?? usually, 15-20 min. for the sterilization step. Clarify.

# C17:0 internal standard, mention the name of the standard

# Lack of more supported references

# Lake of statistical data (data analysis) in the methodology part in addition to results

# Lake of statistical data such as error bars or SD on the figures, tables, and figure legends

# Strain codes should be not in italic

# All tables and figure legends need to reformulate and improve

Author Response

see added file

Reviewer 2 Report

Review of microorganisms-2164738.

What was the key to selecting yeast strains for research?

80-81: the medium has a rather specific composition. On what basis was he chosen? This is not a typical substrate, optimal for growth.

100- ethylacetate:acetonitrile (20:60) - what was the missing 20?

Based on the description of the methodology, I do not understand how the antimicrobial activity was tested, there are no test strains listed.

Figure 1 - no standard deviations. To supplement in all manuscript.

189 – 190 - methodology repetition, please delete

Table 1 - rename  ‘celkové karoteny’ into English

Table 2 - change strain symbols to names - it is inconvenient to interpret

Why is there no strain of Saccharomyces cerevisiae in Figure 3 and in the studies in general? It is described in the methodology, which is very confusing,

Author Response

see added file

Reviewer 3 Report

It would be interesting to add images of the cell lines.

It is necessary to add recent references.

Show in all percentage figures, the error bars.

Figure 7 and 8 should be further discussed.

Author Response

see added file

Round 2

Reviewer 1 Report

The authors addressed most of my comments. 

Reviewer 2 Report

Corrections were made in accordance with the reviewer's comments